# Examining post-conflict stressors in northern Sri Lanka: A qualitative study

**Fiona C. Thomas**[1]\*, **Malasha D'souza**[2], **Olivia Magwood**[3,4], **Dusharani Thilakanathan**[5], **Viththiya Sukumar**[5], **Shannon Doherty**[6], **Giselle Dass**[5], **Tae Hart**[1], **Sambasivamoorthy Sivayokan**[7], **Kolitha Wickramage**[8], **Sivalingam Kirupakaran**[9], **Kelly McShane**[1]

**1** Department of Psychology, Faculty of Arts, Ryerson University, Toronto, Ontario, Canada, **2** Institute for Management and Innovation, University of Toronto, Toronto, Ontario, Canada, **3** CT Lamont Primary Care Research Centre, Bruyère Research Institute, Ottawa, Ontario, Canada, **4** Interdisciplinary School of Health Sciences, Faculty of Health Sciences, University of Ottawa, Ottawa, Ontario, Canada, **5** THEME Institute, Nugegoda, Sri Lanka, **6** School of Allied Health, Faculty of Health, Education, Medicine & Social Care, Anglia Ruskin University, Cambridge, United Kingdom, **7** Jaffna Teaching Hospital, Jaffna, Sri Lanka; Faculty of Medicine, University of Jaffna, Jaffna, Sri Lanka, **8** International Organization for Migration (IOM), United Nations Migration Agency, Migration Health Division, Geneva, Switzerland, **9** Faculty of Graduate Studies, University of Colombo, Colombo, Sri Lanka

\* fiona.thomas@ryerson.ca

**Data Availability Statement:** Data supporting this study are not publicly available because participants did not consent to data sharing and deposit. Please contact Psychology Research Data

## Abstract

Forcibly displaced individuals typically encounter daily stressors, which can negatively impact mental health above and beyond direct exposure to war-related violence, trauma and loss. Understanding the perspectives of war affected communities regarding daily stressors can enhance the integration of mental health into local primary care. The aim of the current study was to explore how daily stressors are conceptualized in a post-conflict setting. Data collection was completed with 53 adult participants who were recruited from primary healthcare clinics in Northern Province, Sri Lanka. Individual interviews were conducted in Tamil, audio-recorded, translated from Tamil to English, and transcribed. Themes emerging from the data were organized into an analytical framework based on iterative coding and grounded in the daily stressors framework. Stressors were conceptualized as chronic stressors and systemic stressors. Findings indicate that chronic stressors, such as loss of property, permeate daily life and have a profound impact on psychological wellbeing. Interviewees additionally reported that systemic stressors stemmed from unresolved grief for missing family members and limited support from institutions. The results of the current study complement existing literature, suggesting the value of multipronged approaches which identify and address symptoms of complicated bereavement while simultaneously alleviating financial hardship. An understanding of stressors experienced by conflict-affected populations in times of chronic adversity can be informative for the design and implementation of culturally-tailored interventions.

at psychresearchdata@ryerson.ca to request for access to the data for the current study.

**Funding:** This research was part funded by the Social Sciences and Humanities Research Council (SSHRC) Vanier Canada Graduate Scholarship awarded to FCT. Additionally, the operational costs of the research project were funded by the 1) Canada Graduate Scholarship Michael Smith Foreign Study Supplement from SSHRC, and 2) Royal Bank of Canada (RBC) Immigrant, Diversity and Inclusion Award from Ryerson University. The granting bodies (i.e., SSHRC, RBC or Ryerson University) did not have a role in study design, data collection and analysis, decision to publish, or preparation of the manuscript.

**Competing interests:** The authors have declared that no competing interests exist.

## Introduction

The Sri Lankan civil war was a 26-year protracted conflict (1983–2009) between the government and the Tamil rebels, notably the Liberation Tigers of Tamil Eelam (LTTE). Consequent to the civil war, approximately 800,000 individuals, families and communities, particularly in the Northern and Eastern Provinces, were uprooted from familiar and traditional ecological contexts within their homeland [1–3]. In these areas, educational and health achievements were significantly impacted, despite relative stability since the end of the civil war in 2009 [4]. The particular circumstances of the conflict in Sri Lanka–prolonged and internal, resulting in displacement, death and disappearances, as well as the tactical use of civilians as hostages–have contributed to severe ramifications to current everyday life [1]. In addition to exposure to extreme war-related traumatic events such as witnessing or experiencing torture, injury and death, many in Sri Lanka have experienced ruptured familial and community relationships, outmigration of loved ones, changing social values, and loss of livelihood and education [5]. As reviewed in a recent World Bank report, many households in the Northern Province turned to daily wage labour as their primary livelihood strategy following the breakdown of critical infrastructure and the agriculture sector, fisheries, and industries during the conflict years [6]. The rise in this informal sector resulted in lower wages for many and increased levels of poverty and displacement in the region [6]. At the end of 2009, half of those displaced were in government supported welfare centres [6]. Patterns of displacement contributed to eroding community and family structures, and weakened social networks and social capital. The resulting economic and social insecurity and vulnerability continues to present day [6].

Several reviews have synthesized the now extensive literature on the topic of mental health of individuals impacted by conflict and forced migration [7–10]. Findings from these reviews suggest that the mental health impact of war-related violence and loss for displaced populations is exacerbated or alleviated depending on the conditions encountered during resettlement and post-conflict; that is, the mental health of communities affected by protracted conflict is powerfully influenced by the *context* following the conflict [11]. Acknowledging the pivotal role of environmental context in mental health is consistent with long-standing social ecological models in psychology [12, 13]. Such frameworks conceptualize human development as rooted in factors at multiple levels such as individual, family, community, and broader society, and across different timepoints (e.g., war exposure, perilous displacement experiences, and post-conflict stressors). The social ecological framework is reflective of the movement in humanitarian settings towards a social determinants of health approach, where trauma-focused interventions are complemented by more holistic psychosocial [14] or public mental health perspectives [15–17].

Specific to populations impacted by displacement, Miller and Rasco proposed a social ecological framework for understanding the extensive range of stressors that may affect the mental health of refugees [15]. In this tradition, distress among displaced communities is understood as stemming not only from war exposure and prolonged violence, but also from the stressful social and material conditions of everyday life that often follow displacement and the destruction of war [11]. The foundational work of Miller and Rasco set the stage for the *daily stressors* framework [11, 18, 19]. Since its introduction in the global mental health (GMH) literature in 2010, the daily stressors framework has been applied extensively in conflict- and post-conflict settings to examine the complex pathways of profound war-related loss, chronic post-conflict contextual difficulties, and the risks for psychological distress [20]. Daily stressors are defined as the chronic and highly stressful social and material conditions that permeate daily life, following protracted, armed conflict [19, 21].

The daily stressors framework has been applied to studies in Sri Lanka to further understand the range of stressors experienced by war affected communities. Much of this literature

is based on studies with children and adolescents [21–24] or conducted with data collected shortly following the end of the conflict [25]. Although there is growing awareness on the social determinants of mental health in general populations [26], this literature is limited when it comes to adults impacted by protracted conflict. To date, literature in the area has not systematically examined the broader social context in a post-conflict situation out of which chronic stressors emerge. Thus, the current study aims to fill these gaps by examining the dynamic relationship between war-related losses and chronic post-conflict stressors, in a population of adults impacted by the protracted conflict in Northern Sri Lanka. For the purposes of this study, these stressors are referred to as *chronic stressors*, to capture both the enduring and noxious nature of the stressors in a post-conflict setting.

## Chronic stressors in Sri Lankan communities affected by protracted conflict

The experience of those displaced is multifaceted, with people across the age spectrum, war widows, ex-combatants, non-combatants and others all facing unique challenges in their post-conflict worlds [27]. Some of these circumstances include protracted violence resulting in the loss of loved ones; breakdown of economic and social infrastructure; loss of social connectedness; and a heightened sense of insecurity [1, 3, 15]. Research has identified widespread individual mental health consequences resulting from war-related violence and loss. A cross-sectional survey conducted in 2011 in the Northwestern Province of Sri Lanka with 450 adults (18–65 years old), revealed a high prevalence of psychopathology (18.8%) among Internally Displaced Persons (IDPs), especially where displacement was prolonged [28].

In addition to epidemiological studies on prevalence rates of psychopathology, emerging literature from contexts of institutionalized violence describe chronic and cumulative socio-economic stressors as integral determinants of mental health [29]. Specifically, individuals experience a great deal more than discrete intermittent events of violence and loss during conflict. In complex situations that follow war and displacement, chronic and stressful social and material conditions resulting from armed conflict additionally permeate daily life [19]. These chronic stressors consist of a range of hardships, related to poor physical health, unemployment, interpersonal violence, social isolation, and barriers to accessing resources like healthcare, education, and vocational services [11, 19].

In Sri Lanka, studies have examined the role of chronic stressors to explain the high rates of psychological distress often identified in war affected communities. For instance, in their study on the effects of mass trauma on family and community systems, Catani et al. drew attention to the need for family and community-based interventions that specifically take into consideration poverty and parental alcohol use in the assessment and treatment of children in post-conflict settings, including in Sri Lanka [22]. In another study, daily stressors were stronger predictors of trauma symptoms than direct exposure to the conflict and the tsunami for youth in Eastern Sri Lanka [23]. The authors concluded that the failure to consider daily stressors in explanatory models of distress among adolescents in emergency settings is likely to overemphasize the predictive power of war and disaster exposure. This could then subsequently neglect important sources of contextual differences that influence youth well-being or distress [23]. Taken together, these studies increase our understanding of the implications of daily stressors in child and youth populations in post-conflict Sri Lanka. However, the impact of stressors on adults affected by the protracted conflict in Northern Sri Lanka remains unclear.

## The current study

As indicated by the literature described above, much of this work on stressors has focused on the experiences of youth and adolescents. Daya Somasundaram has extensively captured and

documented the range of traumatic events and stressors experienced by communities affected by protracted conflict living in the Northern and Eastern Provinces [1, 3, 27]; however, there have not been attempts to categorize the range of these stressors. To date, the understanding and conceptualization of stressors among Sri Lankan communities affected by protracted conflict has primarily focused on *youth chronic stressors* and *adult traumatic stressors*. The current study took an emic approach to understand the types of chronic and systemic stressors experienced among adults affected by the protracted conflict.

This study was conducted in partnership with a five-year program, titled Integrating Mental Health into Primary Care for Post-Conflict Populations in Northern Sri Lanka (COMGAP-S). The aim of COMGAP-S is to enhance the integration of mental health into primary care (see [30] for more details). Building on emerging GMH literature and in partnership with COMGAP-S, the aim of the current qualitative investigation was to explore local perceptions of chronic stressors for communities affected by protracted conflict in Northern Sri Lanka. As noted by Hou et al., an understanding of the chronic, everyday life stressors for displaced populations shifts the focus on environment rather than individual-level interventions, subsequently having reaching impacts for policy and intervention [31].

## Method

### Ethics approval

Ethics approval for the current study was received from Ryerson University in May 2016 and from the University of Jaffna in October 2017. Additional approvals were sought and received from local stakeholders, including the Provincial Director of Health Services in the Northern Province, the Regional Directors of Health in Jaffna and Vavuniya, and finally, physicians from the clinic sites.

**Setting and site selection.** Participants were recruited from primary healthcare clinics (PHC) in two districts in the Northern Province of Sri Lanka, namely Jaffna and Vavuniya, between January and April 2018. Both districts were severely affected by the conflict and the majority of the population was internally displaced at one time or the other, some for as long as 20 years [28] and others for a shorter period of around two years [32]. Established relations with community stakeholders in Jaffna and Vavuniya were present as a result of the COMGAP-S study, which facilitated agreement from local authorities to conduct the current qualitative study. In this way, the sampling design was that of a convenience sample for site determination [33].

Four PHC sites in Jaffna and Vavuniya were selected following consultation with local partners and institutional approvals. Sites included Divisional Hospitals (DHs) and Primary Medical Care Units (PMCUs). In particular, two PMCUs and two DHs were selected in Jaffna and Vavuniya. PMCUs are outpatient care facilities, focused on basic services such as outpatient consultations, dressings, and medication management. DHs are similar to PMCUs with the added integration of inpatient capacity and therefore, often have additional resources [34]. Clinic locations were purposefully selected based on maximum variation sampling [35], ensuring sufficient variability between sites (e.g., size, location within district, geographic reach of patients).

### Sample of participants

Criteria for participation included being internally displaced at any time during the conflict in Sri Lanka and above 18 years of age at the time of the interview. Participants who were unable to provide informed consent due to limited capacity (e.g., significant learning disabilities, hearing difficulties, or those who lacked mental capacity), were not recruited for this study.

Participants were included in the study if impairments did not hinder their ability to provide informed consent and participate. In alignment with the sampling method used for the COM-GAP-S study [30] two local Research Assistants (RAs) approached every fifth adult in the clinic waiting room shortly after they checked in at the front desk. In this way, systematic random sampling was employed [36]. The RAs provided prospective participants with information on the study purpose and approximate time-commitment to participate in the interview. For this study, all participants opted for interviews to be conducted in Tamil. Participants were not remunerated for their participation; however, snacks and beverages were provided as a token of appreciation for interview time. This approach for compensation was recommended follow-ing discussions with the local research team and the University of Jaffna Ethics Review Com-mittee. Interviews continued until saturation was reached. That is, data collection was terminated when a representative sample of participants, inclusive of diversity in age, gender and site, were interviewed and no new insights emerged in interviews [37].

## Procedure

**RA training in interviewing techniques and ethical research procedures.** A qualitative data collection workshop was held in Colombo, Sri Lanka for the RAs and transcriptionist in December 2016. The workshop included direct instruction on ethnographic interviewing tech-niques and ethical research procedures. Training was provided in English, with translation to Tamil with the assistance of a senior RA as needed. A refresher training session was held immediately prior to data collection in December 2018. During the data collection period, eight pilot interviews were conducted in Tamil, audio-recorded, and translated and tran-scribed from Tamil to English. Pilot interviews were important as training alone was insuffi-cient for RAs to feel fully competent in conducting in-depth qualitative interviews.

Audio recordings of interviews were sent to DT, a bilingual transcriptionist (Tamil and English) using a secure file transfer system. Audio-recordings were translated and transcribed from Tamil to English. To ensure accurate translations, a senior bilingual RA (SK) listened to a random sample of eight transcripts (i.e., two from each of the data collection sites) and verified the translations. As the Tamil dialect varies between regions, this was an important step to ensure translations were precise. Translation errors were additionally mitigated by back-checking English transcripts against Tamil audio recordings of pilot interviews and prioritiz-ing substantive over literal meaning. If disagreement between translated words arose (e.g., "field" versus "farm" to describe where food was grown), this was discussed as a team until agreement was reached. Aside from minor word disagreements, there was consensus on the translated transcripts between team members.

**Interview guide and coding approach.** The first author, FCT, drafted a semi-structured interview guide based on an extensive literature review and consultation with experts in the field of daily stressors research. Once FCT developed the interview guide in English, the local RAs translated and back translated it to Tamil. The interview guide was translated and back translated to Sinhalese as per the University of Jaffna ethics application requirements. However, as the Sinhalese population was not the focus of the current study, no Sinhalese participants were recruited, and the Sinhalese interview guide was not implemented. The interview covered several domains including: displacement experience, impact of displacement on daily lives, cur-rent stressors, and constructs related to another study (e.g., coping and support post-conflict).

## Analysis

Data analysis was approached from a phenomenological lens [38, 39]. That is, the goal of the research was to describe the experience of chronic stressors post-conflict in Northern Sri

Lanka. Analysis was conducted on translated, transcribed interviews. Two RAs (MD and OM) assisted with coding and qualitative data analysis. Independently, each researcher initially reviewed six transcripts to gain familiarity with the data. Basic themes were derived from exploring the various topics discussed within the coded segments. In this way, each researcher (FCT, MD and OM) developed a provisional coding framework independently based on the transcripts reviewed. Concepts emerging from the data were then organized into an analytical framework based on iterative coding and grounded in the daily stressors framework [11, 18, 19]. Although the predetermined categories of daily stressors provided a broad conceptual framework, inductive analysis [40] allowed for descriptors to emerge from the data. After this initial step was complete, FCT, MD and OM discussed how their coding aligned or diverged and the themes that emerged. As well, in line with the phenomenological process of coding, the relationship between emerging themes was explored and documented. Thus, the focus was on the *significance*, not the frequency of a code, thereby ensuring that coding was focused on understanding how particular themes connected to others [33].

During the next coding stage, FCT selected six additional transcripts for coding with the revised framework for the three coders in such a way that each transcript was double coded. After this stage was complete, FCT, MD and OM reviewed the coding framework with KM. Additional revisions were made to the coding framework based on analysis of themes and clarifying descriptions of codes. Discrepancies were discussed until consensus was reached on how given segments of text should be coded. To ensure there was not a drift in the definition of codes, FCT developed the coding framework and included definitions for each code, along with examples from the literature and initial coded transcripts. The remaining transcripts were divided between FCT, MD and OM and coded independently. Previously coded transcripts were re-coded based on the final coding framework. Data were coded and analysed with NVivo 12 for Mac, a qualitative software management program.

## Results

In total, 54 participants were recruited. The length of the interviews ranged from 15 minutes (pilot interview) to 80 minutes, with an average of 30 minutes. One interviewee requested to discontinue during the consent process. As such, results from interviews with 53 participants are included in the current study. Participants in Jaffna were aged 32 to 76 years and 78% were female. In Vavuniya, participants were aged 18 to 72 years and 60% were female. Please see Table 1 for participant demographics.

### Social ecological model of stressors

Thematic analysis provided the basis for a conceptual model of stressors. Stressors were conceptualized into two categories–systemic and chronic stressors. Systemic stressors were conceptualized as stressors that are collectively experienced by a population and often, embedded in or impacted by policies and societal infrastructure. Chronic stressors, on the other hand, were understood as stressors experienced at the individual level, typically on a daily basis, and over a number of years.

In line with these conceptualizations, four primary domains of chronic stressors emerged from the data: material loss, personal safety concerns, physical health concerns, and changes to family systems. These stressors occurred in the context of systemic stressors, which included unresolved grief, limited institutional supports, and shifting social values and networks. Fig 1 contains a visual depiction of the Social Ecological Model of Stressors and reflects how themes were organized and conceptualized (e.g., chronic stressors occur in the context of systemic stressors).

**Table 1. Sociodemographic characteristics of participants.**

| Sociodemographic characteristic | Jaffna *n = 23* | | Vavuniya *n = 30* | | Full sample *N = 53* | |
|---|---|---|---|---|---|---|
| | *n* | *%* | *n* | *%* | *n* | *%* |
| Gender | | | | | | |
| Female | 18 | 78.3 | 18 | 60 | 36 | 67.9 |
| Male | 5 | 21.7 | 12 | 40 | 17 | 32.1 |
| Age group | | | | | | |
| 18–34 | 2 | 8.7 | 11 | 36.7 | 13 | 24.5 |
| 35–49 | 6 | 26.1 | 6 | 20.0 | 12 | 22.6 |
| 50–64 | 12 | 52.2 | 7 | 23.3 | 19 | 35.8 |
| 65+ | 3 | 13.0 | 6 | 20.0 | 9 | 17.0 |
| Religion | | | | | | |
| Hindu | 13 | 56.5 | 29 | 96.7 | 42 | 79.2 |
| Christian or Catholic | 10 | 43.5 | 1 | 3.3 | 11 | 20.8 |
| Marital status | | | | | | |
| Married | 21 | 91.3 | 23 | 76.7 | 44 | 83 |
| Never married | 1 | 4.3 | 3 | 10.0 | 4 | 7.5 |
| Widowed, Separated, Divorced or Missing | 1 | 4.3 | 4 | 13.3 | 5 | 9.4 |
| Highest educational level | | | | | | |
| No formal education, other education | 0 | 0.0 | 0 | 0.0 | 0 | 0.0 |
| Grades 1-Grade 5 | 6 | 26.1 | 7 | 23.3 | 13 | 24.5 |
| Grades 6 through O/L[a] | 15 | 65.2 | 19 | 63.3 | 33 | 64.2 |
| University or Higher (including A/L[a]) | 2 | 8.7 | 4 | 13.3 | 6 | 11.3 |
| Employment | | | | | | |
| Employed | 12 | 52.2 | 15 | 50.0 | 27 | 51 |
| Unemployed /disability | 1 | 4.3 | 0 | 0.0 | 1 | 1.9 |
| Student/retired | 1 | 4.3 | 3 | 10.0 | 4 | 7.5 |
| Homemaker [b] | 9 | 39.1 | 12 | 40.0 | 21 | 39.6 |

*Note*. Participants across both sites were on average 49 years old.

[a] O/L = Ordinary Level, approximately equivalent to Grade 11–12 in North American educational standards; A/L = Advanced Level, approximately equivalent to Grade 12 to first year postsecondary education in North American educational standards.

[b] The term 'homemaker' is used to describe participants who noted that they were providing caregiving responsibilities in the home.

**Systemic stressors.** Interviewees identified several systemic stressors, comprised of unresolved grief, limited institutional supports, and shifting social values and networks.

*Unresolved grief.* Almost every participant interviewed for this study was touched by the loss of a loved one during the conflict. Often, participants described a pattern of losing several family members. In this sample, the loss of children was described as particularly difficult to process. In some cases, children became unwell during the conflict and passed away; more commonly however, children were captured during the conflict and their whereabouts remained unknown. This experience has culminated in unresolved grief for many families. Without the ability to engage in burial rituals, some participants talked about maintaining hope for the return of their loved ones, and at the same time, continued to experience grief for their absence:

"My elder son. . .they saw him last in 2009, [but] we still have not seen him. We worry about that. . .that's why I have these chest problems too. I cry whenever I speak about him. We thought he would come back because we heard that they've imprisoned people in

**Fig 1. Social ecological model of stressors.**

various places. An astrologer once told me that he would return by 2017 and that he was mentally affected. So we were expecting him last year but he didn't return [crying]" (Jaffna, 58-year-old female)

As another participant described, even when bodies were returned, the grief persisted as it was not possible to confidently identify bodies. "My younger brother. . .I can't talk about that. That was the biggest loss. . .We still don't know what happened to him. He was tall and big. They said he died and they brought a small coffin. I still don't believe he's dead" (Vavuniya, 49-year- old male). The same participant highlighted the overwhelming grief related to losing

several loved ones: "I lost my relatives. My younger brother, younger sister. . .my brother-in-law died saving others. My uncles, cousins. Who's going to bring them all back?. . .Now I'm all alone. What can I do if this is life?" (Vavuniya, 49-year-old male).

In some cases, female interviewees described the resulting challenges that came with losing a male family member. Compounding unresolved grief, were the implications for safety as well as family income following the passing of (or inability to locate) a loved one. As highlighted by one participant whose brother passed away as a result of shell attacks during the war and whose father was still missing:

> The loss of my father still affects me. We don't know whether he's dead or alive. People even say that we've hid him somewhere to get help. . .When my brother and father were here, they helped manage our home and took care of things financially. Now, we have to do that. We work late and wake up early to get things done so it's very difficult. We were safe when they were around. . .but now we're afraid to leave our house after early evening. . .-Men drink in our village and there are army camps around. These types of situations are unsafe when there are no men in the house (Vavuniya, 25-year-old female)

*Limited institutional supports*. Participants who were displaced to camps reported that they received basic provisions during the conflict such as food aid, basic housing supports, and some medical services in refugee camps, mostly provided by aid or intergovernmental organizations (e.g., International Red Cross, United Nation agencies). Support during the post-conflict situation appeared more fragmented. Some participants reported that the primary support they received from the government or local non-governmental organizations was limited financial support for building new homes and equipment to tend to their land. Others stated that they did not receive any support for housing and instead, built their own shelters. As described by one interviewee, institutional supports were often supplemented with other supports to make ends meet: "Our house was damaged. We now live in a house given by the government, but we've received no other help. When my son came back from abroad, we built a small room extension with his help" (Jaffna, 53-year-old female). Some participants additionally described an inability to depend on institutional supports to obtain what was lost during the war, such as reclaiming their lost properties or searching for missing family members.

*Shifting social values and networks*. Another common theme reported by participants was the impact at a societal level as a result of the protracted nature of the conflict. This impact was at multiple levels, including shifts in social values, politics and systemic changes, and alterations in community settings. In a few instances, participants reported that social circumstances have shifted for the better following the war. For instance, with fewer restrictions following the end of the conflict, participants noted that they were able to reconnect and socialize with neighbours and friends as they previously used to. At the same time, several participants described a pattern of the collectivist culture's demise and that of increased corruption. The following interviewee's statement typified this perception:

> Before, this was a generous place. People used to be invited for a meal if they visited a house. Even though there were problems, there was no scarcity for food. But now, it's difficult for us. Today, poverty is everywhere. . .The government says they are helping people, and they do, but money is stolen by the officials instead of being given to the people. If the government provides officials 1 lakh [100, 000 rupees] to give to people to buy goats, the officials take more than half before distributing the rest. That's life now. . .Nowadays, only some government officials are doing their job honestly. Most others are not. If we try to get work done that requires a government official, they think of it as an opportunity to earn

something [bribes]. Tamils never did that before. . .the dignity that was there before is not here anymore (Vavuniya, 69-year-old male)

The same participant noted how religion has been used for political influence, particularly to foster discrimination: ". . .the religious leaders are behaving in such a way, which is destroying the religion itself. But I don't discriminate others who have different religious beliefs. God is one" (Vavuniya, 69-year-old male). Another participant described the change in community networks, following displacement and relocation of many:

People in our village used to have different habits and behaviours. Most of the people who used to live here aren't around anymore. People from other towns and villages have moved here and captured the land. With our old neighbours, we used to all work together. It's not like that now. The new people don't respect others (Vavuniya, 25-year-old female)

**Chronic stressors.**   In addition to the systemic stressors described above, participants identified a range of chronic stressors experienced following the conflict, including material loss, lack of personal safety, changes to family structure and physical and mental health concerns.

*Material loss*. Participants described immense material loss including their land, homes, modes of transportation (e.g., vehicles, motorcycles), farm animals and equipment, and several other possessions. In addition to losing loved ones, the loss of land and property was the next most salient and frequently cited loss resulting from the conflict. Material loss was often related to several subsequent and interrelated losses. Participants spoke of packing their possessions and taking only what they were able to carry; however, much of their belongings were left behind. Feelings of frustration and sorrow over lost land, homes, and property often surfaced during the interviews.

Everything was lost. . .farm animals, vehicles, [even our] education. I only went with the clothes I was wearing. When we were in the boat, my baby was 45 days old. I just took her, without anything else. I was injured, my son was injured. My husband carried my son, I carried the baby. My chest was wounded. Only when we got to the boat, did they give us some towels and clothes [crying]. We didn't take a single thing from here (Vavuniya, 30-year-old female)

Although institutional supports were provided by the government or non-governmental organizations, participants shared that these provisions were insufficient to cover their needs or to recover what was lost: "Before the war, we worked hard and lived well. We left everything and fled. They only gave us 10% of what we used to have. Recovering the other 90% of what we lost is very difficult, it's impossible" (Vavuniya, 45-year-old male).

Related to loss of material possessions, several participants described the implications of losing their homes and land: "Since we don't have our own land now, we don't have any way of building a house for ourselves. . ..this has affected us" (Jaffna, 59-year-old male). Another frequently cited challenge highlighted by participants was not having the appropriate paperwork to get repossession of their land as a result of fleeing their homes without sufficient notice during the conflict:

In 2008, we were displaced again; we lost all our possessions, and I sent my children away by boat. I couldn't take any of our possessions. . .[When we returned], our land was taken

over by government employees and they sold it. Since we didn't have the deed for our property, the government refused to give back our land. We bought this land. . . I had my shop there. I lived there for 35 years and now it's gone (Vavuniya, 86-year-old male)

At the same time, some participants who were able to access stable housing following the conflict, described the sense of security this brought for them. As noted by one participant, "we plant some things and expect to gain well from that. Before we couldn't do that. We were scared for our lives; we ran here and there with a fear of shell attacks" (Vavuniya, 61-year-old female). The experiences of housing stability were additionally nuanced based on socioeconomic status. One participant who did not have the appropriate paperwork for their home described this discrepancy between those who had the financial means and those who did not: "the ones who have money are living well; they took over land and cut down trees to build houses. But people like us can't get a deed for our land. Thinking about these experiences is painful" (Vavuniya, 40-year-old female).

*Personal safety concerns.* Some participants ironically described feeling more unsafe following the end of the conflict. Safety was reported to be an issue primarily for women and children. As one woman shared, "children, especially girls, are not safe. . .that's why we're scared to stay here now. They kidnap children and shoot young boys. . ." (Jaffna, 76-year-old female). When asked about changes to cultural values and beliefs following the conflict, another participant noted, ". . .the culture has changed a lot. That was one advantage with the LTTE. We were able to go out alone before, even when I was 15 years old. But now we're scared to go out even when we're married" (Vavuniya, 22-year-old female). Some participants attributed the increased risk to personal safety as related to the breakdown of old social connections, because of relocation and changes in community networks.

Other participants described an ongoing fear, despite the end of the conflict: "In 2009 everyone said the conflict was over, but again in 2013 we had issues. They [government] accused us, saying that we were a part of the LTTE. . .I feared for my life" (Jaffna, 40-year-old male). At the same time, a minority of participants echoed a sentiment regarding an increased sense of safety, following the end of the conflict: "[During the conflict], we were always scared whether we were going to lose our children, or if we'll lose our legs [because of landmines]. We were worried for our lives. . .Now, it is better and I don't have these same worries" (Vavuniya, 61-year-old female).

*Changes to family systems.* Participants shared how family structures have shifted following the conflict due to high emigration rates during the conflict and limited economic opportunities for the younger generations. While some participants reported that they felt well-supported by their children or extended family members, selected elderly participants shared their challenges with not being able to turn to their children for financial or emotional support because of concerns of burdening them. Similarly, others described their despondency in being unable to turn to their family members for support given the difficult situations others were also in: "There's not much help [from family]. They are in a difficult situation as well so how could they help?" (Jaffna, 62-year-old female). Other participants, primarily those who were older, described feeling isolated as a result of family moving abroad. Yet, others spoke about the limited employment opportunities for younger generations occasionally resulting in their involvement with alcohol and substance use. One participant explained the implications on her own wellbeing of the aforementioned issues:

I financially supported my grandson to go abroad and I still have that loan pending. I used my gold [to send him abroad] but he came back. . .He barely talks to me now. One other grandson is in prison. It's very difficult to even think [about the situation]. . .someone

introduced him to selling marijuana. The younger generation doesn't know how to earn a living. . .there are so few options (Jaffna, 76-year-old female)

*Physical health concerns.* Participants across the age spectrum discussed their concerns with physical health symptoms, such as diabetes and cardiovascular issues. Some participants recognized that their physical health ailments were related to their chronic stressors: "I'm sick because I'm thinking about our difficulties a lot. Always thinking. . ." (Vavuniya, 86-year-old male). Relatedly, psychosomatic symptoms such as headaches, loss of sleep, and chest pains, were also frequently highlighted by participants.

## Discussion

This study sought to explore how stressors are conceptualized in post-conflict Northern Sri Lanka. Findings from this study underline the multiple dimensions of daily post-conflict life in this context. Four areas of chronic stressors were identified by participants: material loss, personal safety concerns, changes to family systems and physical health concerns following the protracted war. These everyday chronic stressors occurred in the broader context of systemic stressors, namely unresolved grief for missing and deceased family members, limited institutional supports, and changing social values and networks. The data include poignant testimonies of adversity and demonstrate the myriad of ways in which individuals, families and communities respond to chronic stressors.

Participants described material loss and the loss of livelihood as an integral component in an interrelated web of chronic and systemic stressors. Such socioeconomic stressors are reflected in a recent study conducted by the World Bank [6]. In the World Bank study, the authors highlight the intersectional and significant losses of working age males, destruction to infrastructure, and restriction on markets during the conflict years. This combination of losses over several decades continues to be experienced by individuals living in the Northern and Eastern provinces of Sri Lanka.

In the current study, participants shared that material loss extended well beyond the scarcity of basic needs such as food, clothing, and adequate shelter. The loss of property and economic insecurity resulted in complex extensions, where losses in this area perpetuated ongoing losses in other areas, lack of stability in place and income, and honour. Material loss directly conflicted with personal and social aspirations, such as land and home ownership, inability to find suitable marriage proposals for children, and shrinking social networks. In some ways, loss of land appeared symbolically connected to the loss of roots and connection to a particular geographic location, often tied to long-standing social networks, communities, and livelihood. Loss of land also often translated to loss of ancestral homes as properties and land were often passed from one generation to the next. These findings align with studies of other conflict-affected populations. In Mozambique, for example, Sideris found that loss of access to land was one of the worst outcomes of displacement for women refugees [41]. Among IDPs in Northern Uganda, Roberts and colleagues identified that displacement and loss of property resulted in poverty, which in turn adversely impacted psychological wellbeing [42]. Similarly, in their study with children and adult caregivers in Afghanistan, Eggerman and Panter-Brick found that the fundamental concern identified by adult participants was pervasive poverty, which was perceived as the source of all suffering [43]. Similar to participants in the current study, adult interviewees in Afghanistan highlighted their inability to fulfill social obligations, provide stable income for their families, and arrange preferred marriages for their children, all as a consequence of economic insecurity and a "broken economy" [43]. As with

the adult participants in Afghanistan, interviewees in this study voiced the strong relationship between material loss and mental and physical ill health.

In their study with internally displaced women in the Republic of Georgia, Seguin et al. also identified the intrinsic role of the loss of property and livelihood in perpetuating other losses, including social networks and mental and physical health [44]. Drawing on Hobfoll's Conservation of Resources (COR) theory, these cascading losses are referred to as a loss spiral, where losses in one area propagate losses in other areas [45, 46]. The COR theory is a resource-oriented model that proposes that distress is often driven by the loss of important social, psychological and material resources [45, 46]. The COR theory provides another useful framework for understanding the complex ways in which displacement and prolonged violence, can have deleterious and long-standing repercussions for individuals, families, and communities in the current post-conflict context of Northern Sri Lanka. More specifically, the COR theory conceptualizes losses experienced in post-conflict situations as a cascade of losses or loss spirals, whereby large losses–as is often experienced by war affected communities–results in fewer remaining resources to protect against further loss [46].

Although Hobfoll's COR theory was not directly applied in the framework of analysis for this study, participants in the current study described the integral role of material loss in causing or exacerbating a range of other chronic, post-conflict stressors. Exacerbating individual loss, participants described the impact of transgenerational loss across several of the stressors described. For example, several participants referred to their concerns regarding the lack of employment opportunities for their children. This translated to economic strain for families, as adult children typically financially supported their parents. Some participants reflected that lack of employment meant limited prospects for their children's marriage (particularly for sons), which contributed to additional stressors for interviewees. For participants with younger children, sadness and guilt were a common theme at the inability to provide proper education and material items as a result of displacement and loss of property.

Younger participants described the massive disruptions to their education during the conflict, often having to pause and restart their learning as they moved to different camps and towns for safety. These disruptions continued even after resettling, due to the loss of livelihood for their families. The following participant was only able to complete her education until grade 9 due to limited financial means: "we didn't even have money to buy books, we begged our teachers to help us and studied somehow. After a while, we stopped going to school as we couldn't afford to pay tuition" (Vavuniya, 18-year-old female). As noted by Dalgaard and colleagues, the increased levels of loss and suffering in communities affected by protracted conflict have elicited questions in the literature regarding the role of intergenerational trauma transmission [47]. Although it was beyond the scope of this study to explore the impact of loss and suffering across generations, phenomenological analysis highlighted the significance of this finding across all themes. Participants in the current study described experiencing a cascade of losses, not only individually, but also intergenerationally within family systems. Thus, the large cascade of losses as described in Hobfoll's COR theory, results in fewer remaining resources to buffer against future losses *across generations* in this post-conflict context [46].

Grief also emerged as another theme experienced across multiple generations. Specifically, grief for loved ones and complicated bereavement often compounded the deep material losses experienced by war affected communities. The process of grieving can be difficult when survival is of concern, or when other conditions are overwhelming [48]. In their study of Mandaean adult refugees living in Australia who experienced significant loss and mass trauma, Nickerson et al. found that feelings of anger, disappointment, and helplessness were prevalent in those who endorsed symptoms of prolonged grief disorder (PGD) [49]. These findings are similar to what was reported by interviewees in this study. Some respondents additionally

described sudden and violent loss of loved ones during the conflict, resulting in an inability to perform important cultural or religious rituals to symbolize the transitional process of accepting the loss. As highlighted by many participants, this disturbance to the mourning process contributed to difficulty accepting the loss of loved ones, which is similar to the experience described by genocide survivors in Rwanda [50]. In Burundi, participants referred to an 'endless mourning for the dead' or '*ikigandaro*' to describe the grief experienced when their loved ones' bodies were not located [51]. Results from the study in Burundi drew attention to the complicated emotions resulting from the inability to locate the bodies of loved ones, often ensuing in uncertainty [51]. This experience, where individuals oscillated between states of hope and despair, was similar to what many interviewees in the current investigation also reported.

Some female participants described the difficulties that arose following the loss of male family members (e.g., fathers, brothers), especially with regard to economic implications, concerns for personal safety, and increased caregiving responsibilities for surviving members. These findings are in line with what Sivayokan found where families described significant economic downfall following the disappearance of family members, who were often the earning members of the family [52]. In the current study, participants continued to live with the uncertainty of their loved one's death in the case of family members who remained missing. In the literature, this experience has been documented as *ambiguous loss*, which is "a situation of unclear loss resulting from not knowing whether a loved one is dead or alive, absent or present" [53]. The experience of participants in this study are in line with the concept of ambiguous loss and similar to the experience in other post-conflict settings. In Nepal, for example, families of those who disappeared during the conflict likewise described the economic impact of the disappearance of a breadwinner, the difficulty with the lack of closure and inability to engage in end-of-life rituals, associated psychological symptoms, and the need for constructing meaning from the loss [54]. Somasundaram and Sivayokan describe in detail the consequences of the disappearance of loved ones in Sri Lanka and note the following: "the after-effects of disappearance are enormous and can affect individuals, families, communities and the society at large. . .It is an ongoing, everlasting, never-ending stressor with waxing and waning effect" [55].

## Limitations

During initial interviews in Jaffna, it was challenging to recruit participants in a younger age demographic. Only two participants (8.7% of Jaffna sample) identified in the 18 to 34 age group in Jaffna. The low proportion of participants who identified in this category may have been a result of the time-of-day data collection was conducted (mornings and afternoons). The participating PHC sites were not open in the evenings. It is possible that younger adults who worked outside of the home during the day were inadvertently excluded and thus, underrepresented in the sample from Jaffna. RAs were provided additional training to increase recruitment of participants across the age spectrum. To gather sufficient information and varied perspectives, additional participants were recruited in Vavuniya. In Vavuniya, 11 participants (36.7% of Vavuniya sample) identified in the 18–34 age group. The degree of saturation achieved on the themes suggest that the major stressors described by this group of participants were covered.

Relatedly, as indicated in Table 1, 43.5% of participants in Jaffna identified as Christian or Catholic. Based on 2012 census data, 12.5% of the population in Jaffna identified as Roman Catholic, which is the largest endorsed religion after Hinduism in the region [56]. Further disaggregation for those who identify as Christian is not indicated in the 2012 census [56]. Although we made significant efforts to collect data from a representative sample (e.g.,

maximum variation sampling for sites, systematic random sampling for participants), our sampling design was that of a convenience sample for site determination. Thus, the selection of clinics where data was collected may have influenced the high number of participants in our sample who identified themselves as Roman Catholic/Christian in Jaffna. Of note however, majority of the participants in the current study identified as Hindu (79.2%). Future studies may benefit from further stratified random sampling of participants who identify as Hindu and other religions to explore any differences in how religion may intersect with chronic stressors experienced.

Additionally, interviews were conducted at a single time point, which may have been insufficient time to develop trust and enable participants to openly discuss issues related to their daily lives. Participants also typically travelled by public transportation to access medical services and may not have been inclined to spend significant time in the interview. As an outsider in this context, it is important to recognize how FCT's presence during the data collection process may have influenced what was shared by participants. To circumvent such issues, the first author was not physically in the same room for any interviews.

Of additional consideration is whether interviewees were reluctant to share in-depth narratives out of fear. Several researchers have elaborated on a 'culture of silence' [57, 58], which has been imposed in Northern Sri Lanka through indoctrination, intimidation, abductions, extrajudicial killings, and torture [27]. This perspective was shared anecdotally by members of the local research team. Similar observations were made by Sivayokan and Somasundaram when discussing the psychological impact of fear and is exemplified in the following quote: "you see, we are all terrified. We will not open our mouths for anything. We will survive only by being silent. I tell you all this because I know you well" [55]. That being said, as RAs became more comfortable with applying important interviewing techniques (e.g., asking for elaboration, refraining from the use of leading questions, probing further by asking difficult questions), interview length increased suggesting participants felt comfortable discussing their displacement history and current circumstances.

## Implications and conclusion

Despite the aforementioned limitations, the current study gives qualitative depth to epidemiological and quantitative studies demonstrating the impact of chronic stressors on communities affected by the protracted conflict in Northern Sri Lanka. The findings additionally contribute to previous literature on the impact of protracted war on communities. This study has several implications. As described by Wells and colleagues based on their work with Syrian refugees in Jordan, "an ecological approach goes beyond a cultural formulation which adapts imported therapeutic models to local explanatory models [59]. Rather, an ecological model calls for a formulation which considers the interplay of social and cultural factors" [60]. The current study proffers such a social ecological model, which encapsulates the contextualized experience of loss and stressors for communities affected by the protracted conflict in Northern Sri Lanka. The current findings support existing literature that emphasizes the integration of mental health supports with layered services, most notably the Inter-Agency Standing Committee (IASC) Guidelines on Mental Health and Psychosocial Support in Emergency [14]. Miller and Rasmussen similarly call for integrated interventions that address both the chronic and war-related stressors in a sequential approach [18]. In Sri Lanka, Somasundaram and Sivayokan call for a community based, public mental health approach [5]. In particular, they highlight the importance of interventions targeted at advancing socioeconomic opportunities, the integration of cultural rituals and ceremonies, and acknowledging the importance of the ancestral home and village–all findings that continue to be echoed by participants in the current study.

Economic stress in the context of bereavement is an understudied area; further research in this area can inform the development and implementation of targeted interventions focused on simultaneously addressing chronic and war-related stressors. The results of the current study suggest the value of multipronged approaches that seek to not only identify and address symptoms of complicated bereavement, but also couple such interventions with alleviating financial hardship. Services, targeted in tandem at addressing chronic stressors while assessing the need for specialized clinical services, are certainly one effective approach for clinical interventions [61]. It is possible that psychological assistance will be necessary for some individuals who are substantially impaired by complicated bereavement before they are able to take advantage of programs aimed at reducing financial hardship [11]. To address the interrelated and social determinants of mental health in post-conflict settings, intersectoral and interdisciplinary action is called for [62].

Specific to the Northern Sri Lankan context, interventions aimed at alleviating both financial hardship and processing complicated bereavement would require expertise from the social welfare sector, clinicians, as well as those with expertise in cultural rituals and practices related to grief. Clinical intervention design can benefit from the integration of culturally mediated protective mechanisms, such as rituals and ceremonies. In Eastern Sri Lanka, for example, Lawrence describes the practice of '*vakku choluthal*', particularly in situations of disappearance or ambiguous loss [63]. In this ritual, families are informed of what happened to the disappeared loved one in a socially supportive context. *Koothu*, a form of traditional, folk drama that is performed with music and dance, is documented in the literature as another mechanism of processing grief in post-conflict Sri Lanka [5]. Such traditional coping strategies were not cited by participants in the current study. It is possible that during the conflict and in the immediate aftermath of its conclusion, individuals were more inclined to engage in such community grieving rituals. With the passage of time and focus on postwar recuperation, such rituals may have become less of a priority for war affected communities in the current study. Nonetheless, as indicated by the results here, many participants continue to suffer from the uncertainty of ambiguous loss and complicated bereavement in situations of sudden and traumatic loss. Further research to determine the effectiveness of such integrated interventions are warranted.

Similarly, researchers have called for an in-depth understanding of the relational sequelae of distress, rather than intervention modalities that prioritize an individualizing approach [64]. Such calls for action are supported by the current study findings, particularly with regard to the common thread of intergenerational loss and suffering that emerged across several themes in the current study. In other words, findings from the current study highlight the importance of theoretical approaches that recognize contextual stressors subsequently leading to clinical interventions that may have more utility. Specifically, the results of this study underscore the importance of identifying the dynamic interplay between the individual, family, and their environment. Finally, it is only by privileging community knowledge that a new generation of interventions that are contextually grounded, intersectoral, and reflective of the priorities of communities themselves, can begin to emerge.

## Acknowledgments

The authors gratefully acknowledge and thank the following study, research, policy, and clinical staff for their commitment and support of this study: Perumal Menuja, Rajendra Surenthirakumaran, the Provincial Director of Health Services in the Northern Province, the Regional Directors of Health in Jaffna and Vavuniya, and finally, the physicians and healthcare staff from the participating study sites. We also thank the study participants for their interest,

partnership, and contributions to the current study. Finally, we are indebted to Dr. Chesmal Siriwardhana (1978–2017), for without him, none of this work would have been possible. Thank you for your unwavering support, dedication as a mentor, colleague and an ethical scholar, and above all, for modeling the true meaning of collaborative research.

## Author Contributions

**Conceptualization:** Fiona C. Thomas, Shannon Doherty, Giselle Dass, Tae Hart, Sambasivamoorthy Sivayokan, Kolitha Wickramage, Kelly McShane.

**Data curation:** Fiona C. Thomas, Dusharani Thilakanathan, Viththiya Sukumar, Sivalingam Kirupakaran.

**Formal analysis:** Fiona C. Thomas, Malasha D'souza, Olivia Magwood, Kelly McShane.

**Funding acquisition:** Fiona C. Thomas, Sambasivamoorthy Sivayokan, Kelly McShane.

**Investigation:** Fiona C. Thomas, Tae Hart, Kelly McShane.

**Methodology:** Fiona C. Thomas, Tae Hart, Sivalingam Kirupakaran, Kelly McShane.

**Project administration:** Fiona C. Thomas, Shannon Doherty, Giselle Dass, Sivalingam Kirupakaran.

**Resources:** Fiona C. Thomas, Shannon Doherty, Giselle Dass, Sambasivamoorthy Sivayokan, Sivalingam Kirupakaran, Kelly McShane.

**Supervision:** Fiona C. Thomas, Shannon Doherty, Tae Hart, Sambasivamoorthy Sivayokan, Kolitha Wickramage, Sivalingam Kirupakaran, Kelly McShane.

**Validation:** Fiona C. Thomas, Tae Hart, Sambasivamoorthy Sivayokan, Kelly McShane.

**Writing – original draft:** Fiona C. Thomas, Sivalingam Kirupakaran.

**Writing – review & editing:** Fiona C. Thomas, Malasha D'souza, Olivia Magwood, Shannon Doherty, Giselle Dass, Tae Hart, Sambasivamoorthy Sivayokan, Kolitha Wickramage, Sivalingam Kirupakaran, Kelly McShane.

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
