## [Decision Letter · Decision Letter 0]

23 Dec 2021

PONE-D-21-25025Examining Post-Conflict Stressors in Northern Sri Lanka: A Qualitative StudyPLOS ONE

Dear Dr. Thomas,

Thank you for submitting your manuscript to PLOS ONE. After careful consideration, we feel that it has merit but does not fully meet PLOS ONE’s publication criteria as it currently stands. Therefore, we invite you to submit a revised version of the manuscript that addresses the points raised during the review process.

We look forward to receiving your revised manuscript.

Kind regards,

Johnson Chun-Sing Cheung, D.S.W.

Academic Editor

PLOS ONE

**Comments to the Author**

Reviewer #1: This study investigates examines the post-conflict stressors in northern Sri Lanka. For this purpose, the authors conducted qualitative interview with 53 adult participants who were affected by the Sri Lankan civil war. The results reveal substantial chronic stressors such as loss of property as well as systemic stressors stemmed from unresolved grief and limited support from institution.

This is an important study that analyzes mental well-being among those who are severely affected by a civil conflict. The interview was conducted with great care, and the results are highly reliable.

Major Comments:

(1) A more detailed discussion of the representativeness of the sample is needed. For example, according to Table 1, 43.5% of the participants are Christian or Catholic in Jaffna, which seems quite high comparing to the census results. Does this mean the sample is not representative? If it reflects the reality, what is the mechanism of this higher proportion? Do they convert after the civil war?

(2) Related to (1), the description of the participants’ socio-economic status (e.g., income, asset, etc…) comparing to non-affected people is necessary to understand their current situation. Furthermore, the change in socio-economic status between pre- and post-conflict would also be informative.

(3) Since the authors categories the systemic and chronic stressors, it would be informative to prepare a table showing how many of the participants referred to each category.

Reviewer #2: This well-written manuscript focuses on the impact that chronic and systemic stressors have on survivors of the Sri Lankan civil war. I was very impressed by this manuscript, which stands as an example of what quality qualitative research in the social sciences can provide, i.e., a grounded, carefully observed, detailed description and analysis of complex situations with a representative (enough) sample. While our understanding of war-affected populations has developed a great deal in the last 25 years or so, there is still a dearth of research that focuses on the roles that context and culture play in impacting the mental well-being of survivors of war living in post-war settings. This paper is an example of the kind of research that is required in order for us to better understand the determinants of mental distress in these populations, so we can them develop effective interventions.

The aspect of the study described in this manuscript that I was especially impressed by was the sampling method. Many studies conducted in non-Western war-affected populations rely on convenience samples; this was not the case here. The authors made a significant effort to collect data from a representative sample in North-East Sri Lanka; while they weren't able to fully achieve this (the number of younger adults in their sample was low compared to the population they were sampling from), the result of their efforts was a sample that was a great deal more representative of the population they were studying than most other qualitative studies of war-affected non-Western populations.

---

## [Author Response · Author response to Decision Letter 0]

21 Mar 2022

Dear Dr. Cheung and reviewers of PLOS One:

Thank you for your helpful feedback on our manuscript entitled, Examining post-conflict stressors in Northern Sri Lanka: A qualitative study. We understand this was a competitive process and are grateful for the opportunity to revise our manuscript and respond to your comments. 

We have endeavored to address the questions and comments raised by the two reviewers. We have made changes within the manuscript in track changes as requested. A clean version of the manuscript is also attached. In addition, we include responses to specific feedback from the reviewers below.

Thank you again for your time in reviewing this manuscript and providing us with feedback to strengthen our article.

Sincerely,

Fiona C. Thomas*, Ph.D.,

Department of Psychology, Ryerson University

350 Victoria Street, Toronto, Ontario, Canada M5B 2K3

Phone: 1-416-898-4743; E-Mail: fiona.thomas@ryerson.ca

& co-authors:

Malasha D’souza; Olivia Magwood; Dusharani Thilakanathan; Viththiya Sukumar; Shannon Doherty; Giselle Dass; Tae Hart; Sambasivamoorthy Sivayokan; Kolitha Wickramage; Chesmal Siriwardhana; Sivalingam Kirupakaran; Kelly McShane

Editor Feedback 

Response: We have formatted the manuscript to meet PLOS ONE's style requirements. Please note that we have revised the image for Figure 1. The revised image has also been saved and uploaded with the appropriate file name. 

Response: Thank you for alerting us to this error. We have now made the appropriate changes. Please note that an award number is unavailable for the following funding source: Royal Bank of Canada (Immigrant, Diversity and Inclusion Award).

Response: 

As researchers, we are committed to open access, and we would like to make our minimal data set fully available for use by other researchers. As it stands however, we are unable to provide our study’s minimal underlying data set as either Supporting Information files or to a stable, public repository. Below we outline the process we used to reach this decision.

Initially, we reviewed the policies at our local institution and within our country. As per the Government of Canada's Guidance on Depositing Existing Data in Public Repositories, there is clear indication that we need consent: (1) from either the participants or (2) the Research Ethics Board to deposit in public repositories. As for the first option, during data collection, we did not specifically request consent from participants to include their data in a public repository. An alternative would be to go back to the individual participants and ask them if they would consent now. However, it is not possible to request consent from participants as we did not gather their contact information at the point of data collection (in 2018). The team had decided it was not necessary to collect contact information as it was not needed for the purposes of our research question. Thus, it would be impossible to contact all participants for their consent. With respect to the second option, it is also impractical to request consent from the Research Ethics Boards (REB) as the current study was granted approval from two institutions in 2018, in two different countries (i.e., Ryerson University and the University of Jaffna). We are concerned that making this request for additional consent to deposit the data in a repository will delay the dissemination of the current manuscript. In addition, we also believe that it would be preferable for participants themselves to grant consent for their own data given the nature of this study (e.g., experiences in the transition post-conflict), as well the fact that the data is qualitative in nature. Accordingly, we have reached the conclusion that it is not possible to provide our study’s minimal underlying data set as either Supporting Information files or to a stable, public repository.

Moving forward, we will be mindful of requesting explicit consent from participants to include their data in a public repository for future studies. As indicated in the Data Availability Statement below, we remain open to sharing our data for research purposes with specific individuals and institutions if they contact us at the email address below. As noted by PLOS ONE requirements, we have ensured that the request for access to the data can be made to an institution, rather than an individual.

Suggested Data Availability Statement: Data supporting this study are not publicly available because participants did not consent to data sharing and deposit. Please contact Psychology Research Data at psychresearchdata@ryerson.ca to request for access to the data for the current study. 

Response:

The references have been updated throughout to give the most recent data available. Additionally, references have been formatted to Vancouver Style as per PLOS ONE requirements. No references were retracted but the following references were added to address reviewer feedback: 

O’Donnell A, Ghani Razaak M, Kostner M, Perumpillai-Essex J. Shadows of Conflict in Northern and Eastern Sri Lanka: Socioeconomic Challenges and a Way Forward [Internet]. Washington, DC: World Bank; 2018 [cited 2022 Feb 11]. Available from: https://openknowledge.worldbank.org/handle/10986/30545

Department of Census and Statistics (Sri Lanka), United Nations Development Programme (UNDP), United Nations Population Fund (UNFPA). Sri Lanka Population and Housing Census 2012 [Internet]. Sri Lanka; 2012 [cited 2022 Feb 11]. Available from: http://ghdx.healthdata.org/record/sri-lanka-population-and-housing-census-2012

Comments to the Author

Reviewer #1: This study investigates examines the post-conflict stressors in northern Sri Lanka. For this purpose, the authors conducted qualitative interview with 53 adult participants who were affected by the Sri Lankan civil war. The results reveal substantial chronic stressors such as loss of property as well as systemic stressors stemmed from unresolved grief and limited support from institution.

This is an important study that analyzes mental well-being among those who are severely affected by a civil conflict. The interview was conducted with great care, and the results are highly reliable.

Major Comments:

1. A more detailed discussion of the representativeness of the sample is needed. For example, according to Table 1, 43.5% of the participants are Christian or Catholic in Jaffna, which seems quite high comparing to the census results. Does this mean the sample is not representative? If it reflects the reality, what is the mechanism of this higher proportion? Do they convert after the civil war?

Response:

Thank you for this important observation regarding the high proportion of participants who identify as Christian or Catholic. My co-authors, particularly those who are locally based and/or working in northern Sri Lanka provided their input into potential reasons for the high percentage of Christian or Catholic participants in Jaffna. From our research, we identified the following information. 

The last Census of Population and Housing was conducted in 2012. The prior census was conducted on July 17th, 2001. However, no enumeration was conducted for Jaffna, Kilinochchi and Mullaitivu districts. Enumeration was partially gathered for Mannar and Vavuniya districts in Northern province. Prior to 2001, there was a gap of two decades where census data was not collected (for further information, http://www.statistics.gov.lk/about_us/history). The limited availability of census data makes inferences on the change in demographics as pertaining to religion challenging.

That being said, data from the 2012 census indicates that 12.9% of the population in Jaffna identified as Roman Catholic (disaggregation by Christian and Catholic is not provided), which is the largest endorsed religion after Hinduism in the region. 

In Jaffna, data for the current study was collected in the towns of llavalai and Urumprai. Illavalai is in Sandilipay DS and Urumprai is in Kopay DS. Based on the 2012 census data, 20.3% of the population in Sandilipay DS identify as Roman Catholic and 5.6% of the population in Kopay DS identify as Roman Catholic. These numbers are indeed lower than the percentage of participants in our sample who identify as Roman Catholic in Jaffna. Although we made significant efforts to collect data from a representative sample (e.g., maximum variation sampling for sites, systematic random sampling for participants), our sampling design was that of a convenience sample for site determination and thus, the selection of clinics where data was collected may have influenced the high number of participants in our sample who identified themselves as Roman Catholic/Christian. Of note however, majority of the participants in the current study identified as Hindu (79.2%).

Nonetheless, we have acknowledged this as a limitation. 

2. Related to (1), the description of the participants’ socio-economic status (e.g., income, asset, etc…) comparing to non-affected people is necessary to understand their current situation. Furthermore, the change in socio-economic status between pre- and post-conflict would also be informative.

Response:

Thank you for this recommendation. For the purposes of the current study, we did not collect data from non-affected people in Sri Lanka as that did not align with the research question. Additionally, we did not explicitly ask how their socioeconomic status changed between pre- and post-conflict, nor was this data gathered as part of the larger parent study either (COMGAP-S). However, in reviewing the literature, we identified some sources that attempt to explore questions related to socio-economic status. These are reviewed and briefly discussed in the manuscript (i.e., pgs 3; 22-23). 

3. Since the authors categories the systemic and chronic stressors, it would be informative to prepare a table showing how many of the participants referred to each category.

Response:

Thank you for this feedback as it prompted us to consider our lens. Your question brought to the forefront the distinction between a positivist lens where we look to quantify the qualitative themes versus the phenomenological lens which was applied in our analysis. The phenomenological lens is premised on describing the lived experience of participants based on the intersection and nuances of themes that emerged. In this way, we recognize that the themes identified are based on the salience and intensity of narratives shared, rather than the frequency as suggested by the positivist lens. 

Based on the feedback provided here, I consulted with an expert in qualitative research methods in the Department of Psychology at Ryerson University. Consistent with the phenomenological lens to qualitative analysis, the recommendation was to avoid quantifying the number of participants who endorsed each theme. The frequency of themes endorsed will overlap with the themes identified in the study based on their salience and intensity versus frequency as suggested by the positivist lens. 

In follow up to your insightful feedback, we have carefully re-reviewed the manuscript to ensure the themes noted indeed reflect the intensity, salience, depth as is typical of the phenomenological lens and are confident that is indeed the case based on the analytical procedure followed. 

Reviewer #2: This well-written manuscript focuses on the impact that chronic and systemic stressors have on survivors of the Sri Lankan civil war. I was very impressed by this manuscript, which stands as an example of what quality qualitative research in the social sciences can provide, i.e., a grounded, carefully observed, detailed description and analysis of complex situations with a representative (enough) sample. While our understanding of war-affected populations has developed a great deal in the last 25 years or so, there is still a dearth of research that focuses on the roles that context and culture play in impacting the mental well-being of survivors of war living in post-war settings. This paper is an example of the kind of research that is required in order for us to better understand the determinants of mental distress in these populations, so we can them develop effective interventions.

The aspect of the study described in this manuscript that I was especially impressed by was the sampling method. Many studies conducted in non-Western war-affected populations rely on convenience samples; this was not the case here. The authors made a significant effort to collect data from a representative sample in North-East Sri Lanka; while they weren't able to fully achieve this (the number of younger adults in their sample was low compared to the population they were sampling from), the result of their efforts was a sample that was a great deal more representative of the population they were studying than most other qualitative studies of war-affected non-Western populations.

Response: Thank you for this positive feedback!

---

## [Decision Letter · Decision Letter 1]

1 Apr 2022

Examining Post-Conflict Stressors in Northern Sri Lanka: A Qualitative Study

PONE-D-21-25025R1

Dear Dr. Thomas,

We’re pleased to inform you that your manuscript has been judged scientifically suitable for publication and will be formally accepted for publication once it meets all outstanding technical requirements.

Kind regards,

Johnson Chun-Sing Cheung, D.S.W.

Section Editor

PLOS ONE

---

## [Editor Report · Acceptance letter]

23 Aug 2022

PONE-D-21-25025R1 

Examining post-conflict stressors in northern Sri Lanka: a qualitative study 

Dear Dr. Thomas:

I'm pleased to inform you that your manuscript has been deemed suitable for publication in PLOS ONE. Congratulations! Your manuscript is now with our production department. 

Kind regards, 

on behalf of

Dr. Johnson Chun-Sing Cheung 

Section Editor

PLOS ONE